# PROBABILISTIC ADAPTATION OF BLACK-BOX TEXT-TO-VIDEO MODELS

**Sherry Yang**[*,1,2], **Yilun Du**[*,3], **Bo Dai**[1],
**Dale Schuurmans**[1,4], **Joshua B. Tenenbaum**[3], **Pieter Abbeel**[2]
[1]Google DeepMind, [2]UC Berkeley, [3]MIT, [4]University of Alberta
video-adapter.github.io

## ABSTRACT

Large text-to-video models trained on internet-scale data have demonstrated exceptional capabilities in generating high-fidelity videos from arbitrary textual descriptions. However, similar to proprietary language models, large text-to-video models are often black boxes whose weight parameters are not publicly available, posing a significant challenge to adapting these models to specific domains such as robotics, animation, and personalized stylization. Inspired by how a large language model can be prompted to perform new tasks without access to the model weights, we investigate how to adapt a black-box pretrained text-to-video model to a variety of downstream domains without weight access to the pretrained model. In answering this question, we propose *Video Adapter*, which leverages the score function of a large pretrained video diffusion model as a probabilistic prior to guide the generation of a task-specific small video model. Our experiments show that, by incorporating broad knowledge and fidelity of the pretrained model probabilistically, a small model with as few as 1.25% parameters of the pretrained model can generate high-quality yet domain-specific videos for a variety of downstream domains such as animation, egocentric modeling, and modeling of simulated and real-world robotics data. As large text-to-video models are starting to become available as a service, similar to large language models, we advocate for private enterprises to expose the scores of video diffusion models as outputs in addition to generated videos to allow flexible adaptation of large pretrained text-to-video models by the general public. See website at https://video-adapter.github.io.

## 1 INTRODUCTION

Large text-to-video models with billions of parameters trained on internet-scale data have become capable of generating highly realistic videos from general text descriptions (Ho et al., 2022; Hong et al., 2022; Singer et al., 2022). When models are used for specific domains such as generating video plans for robotics (Du et al., 2023b) and self-driving cars (Santana & Hotz, 2016), videos for animation (Wang et al., 2019), or videos with customizable styles similar to those common in text-to-image (Wang et al., 2019; Ramesh et al., 2022; Liu et al., 2022; Gal et al., 2022; Ruiz et al., 2022; Zhang & Agrawala, 2023), a pretrained text-to-video model requires task-specific adaptation. Efficient and effective adaptation of text-to-video models is what stands in the way from expanding their current application of these models in media and entertainment to their potential to solve real-world problems by modeling real-world physics and dynamics in problem-specific settings.

Unfortunately, similar to state-of-the-art language models (OpenAI, 2023; Anil et al., 2023), pretrained text-to-video models are black boxes to the general public; one can use them to generate videos, but not finetune them to solve domain-specific tasks, as the parameters of pretrained text-to-video models are not publicly available (Ho et al., 2022; Villegas et al., 2022; Blattmann et al., 2023). This rules out direct applications of efficient finetuning from text-to-image, such as LoRA in stable diffusion (Smith et al., 2023), DreamBooth (Ruiz et al., 2023), and ControlNet (Zhang & Agrawala, 2023), which require access to the pretrained model weights. While some finetuning-free techniques can control image generation by manipulating visual or textual features (Ramesh et al., 2022; Gal et al., 2022), it is not clear how to manipulate these features for video generation, as these features

---

[*]Equal contribution. Correspondence to: Sherry Yang <sherryy@berkeley.edu>, Yilun Du <yilundu@mit.edu>.

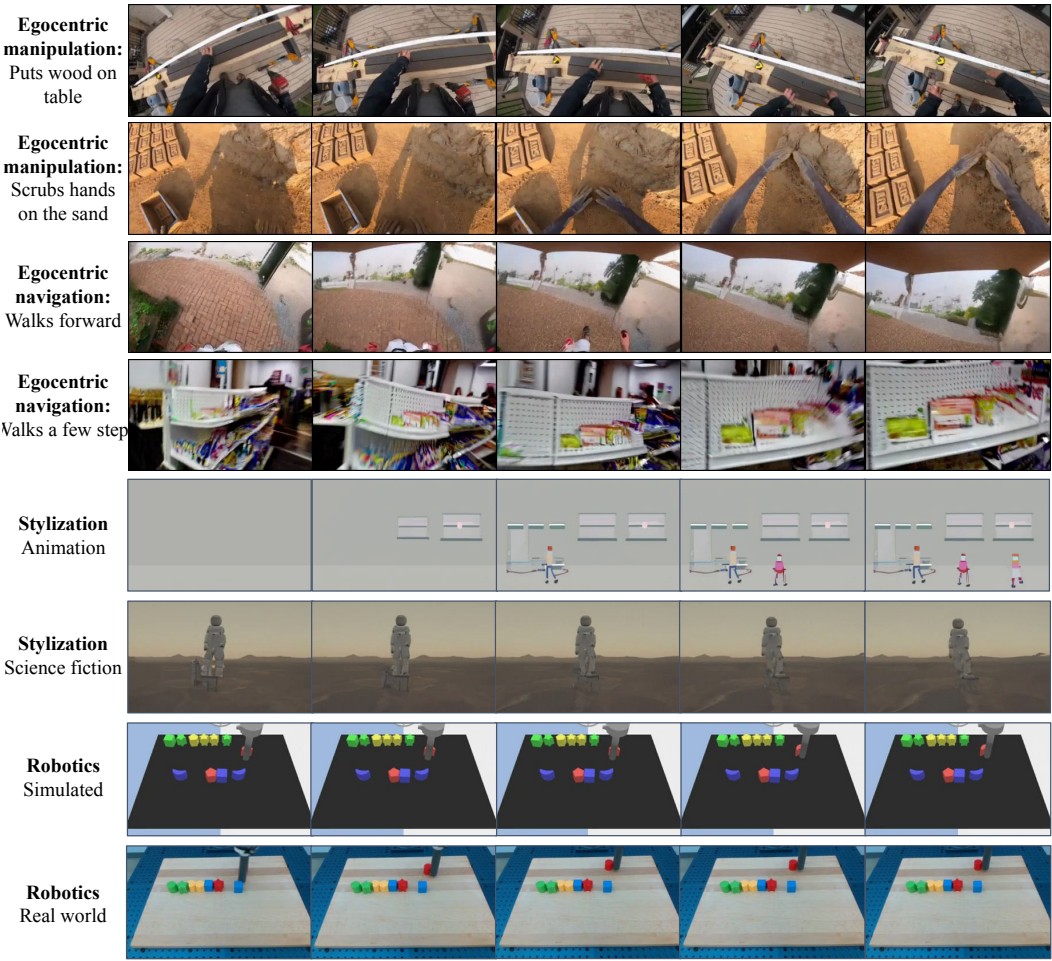

Figure 1: **Video Adapter Generated Videos.** Video Adapter is capable of flexible generation of diverse videos with distinct styles including videos with manipulation and navigation based egocentric motions, videos with personalized styles such as animation and science fictions, and simulated and real robotic videos.

would then have to capture complex temporal information extracted from networks with orders of magnitude larger sizes than text-to-image (Blattmann et al., 2023).

Inspired by finetuning-free adaptation of language models through in-context learning (Brown et al., 2020) and sophisticated prompting (Wei et al., 2022), which essentially modify the prior distribution of pretrained langauge models to perform specific tasks, we ask the natural question of whether it is possible to modify the prior distribution of pretrained text-to-video models to achieve downstream tasks without finetuning the pretrained model. Intuitively, even though the exact video statistics in a downstream task differ from the pretraining videos, certain video properties such as dynamics of the world and semantics of objects from the large pretrained model are still tremendously helpful to the generation of downstream videos. This suggests that a large pretrained video model could be used as a knowledge prior to guide the generation of task-specific videos while maintaining broad properties such as temporal consistency and object permanence.

To this end, we propose Video Adapter, a probabilistic approach for exploiting a black-box video diffusion model to guide the generation of task and domain specific videos. By factoring the domain-specific video distribution into a pretrained prior and a small trainable component, we can preserve desirable characteristics of the pretrained model (i.e., temporal consistency and object permanence) in generating specialized videos, effectively adapting the black-box pretrained model without requiring access to the pretrained model weights. One limitation of Video Adapter is that it requires scores of the black-box video diffusion model as outputs, but we note that this is hard to avoid if one wants to effectively use the broad knowledge of pretrained models. Therefore, we advocate for proprietary

text-to-video APIs to expose diffusion scores as additional outputs to broaden the applications of large video diffusion models.

We evaluate Video Adapter on a variety of tasks and domains as illustrated in Figure 1. Quantitatively, Video Adapter achieves better FVD and Inception Scores than the pretrained video model or the task-specific small models in generating domain specific videos for robotics (Ebert et al., 2021) and egocentric movements (Grauman et al., 2022). Qualitatively, we show that Video Adapter can generate stylized videos such as sci-fi and animation, and further enable domain randomization in robotics (Tobin et al., 2017) for bridging sim-to-real (Zhao et al., 2020) through randomized stylisation of lighting and distractors.

## 2 PRELIMINARIES

We first introduce relevant background information on denoising diffusion probabilistic models (DDPMs) and discuss their connection to Energy-Based Models (EBMs). We will then use this connection to EBMs to convert black-box video diffusion models to probabilistic priors.

**Denoising Diffusion Probabilistic Models.** Denoising diffusion probablistic models (Sohl-Dickstein et al., 2015; Ho et al., 2020) are a class of probablistic generative models where the generation of a video $\tau = [x_1, \ldots, x_H] \in X^H$ is formed by iterative denoising. Given a video $\tau$ sampled from a video distribution $p(\tau)$, a randomly sampled Gaussian noise variable $\epsilon \sim \mathcal{N}(\mathbf{0}, \mathbf{I})$, and a set of $T$ different noise levels $\beta_t$, a denoising model $\epsilon_\theta$ is trained to denoise the noise corrupted video $\tau$ at each specified noise level $t \in [1, T]$:

$$\mathcal{L}_{\text{MSE}} = \left\| \epsilon - \epsilon_\theta \left( \sqrt{1 - \beta_t} \tau + \sqrt{\beta_t} \epsilon, \, t \right) \right\|^2$$

Given this learned denoising function, new videos may be generated from the diffusion model by initializing a video sample $\tau_T$ at noise level $T$ from a Gaussian $\mathcal{N}(\mathbf{0}, \mathbf{I})$. This sample $\tau_T$ is then iteratively denoised following the expression:

$$\tau_{t-1} = \alpha_t(\tau_t - \gamma_t \epsilon_\theta(\tau_t, t) + \xi), \quad \xi \sim \mathcal{N}(\mathbf{0}, \sigma_t^2 \mathbf{I}), \tag{1}$$

where $\gamma_t$ is the step size of denoising, $\alpha_t$ is a linear decay on the currently denoised sample, and $\sigma_t$ is a time varying noise that depends on $\alpha_t$ and $\gamma_t$. The final sample $\tau_0$ after $T$ rounds of denoising corresponds to the final generated video.

**Energy-Based Models View of DDPMs.** The denoising function $\epsilon_\theta$ estimates the score (Vincent, 2011; Song & Ermon, 2019; Liu et al., 2022) of an underlying (unnormalized) EBM probability distribution (LeCun et al., 2006; Du & Mordatch, 2019) characterizing the noise perturbed data. Therefore, a diffusion model corresponds to an EBM, $p_\theta(\tau) \propto e^{-E_\theta(\tau)}$, where the denoising function is given by $\epsilon(\tau_t, t) = \nabla_\tau E_\theta(\tau_t)$. The sampling procedure in a diffusion model corresponds to the Langevin sampling procedure on an EBM (see derivation in Appendix A):

$$\tau_{t-1} = \alpha_t(\tau_t - \gamma \nabla_\tau E_\theta(\tau_t) + \xi), \quad \xi \sim \mathcal{N}(\mathbf{0}, \sigma_t^2 \mathbf{I}). \tag{2}$$

This equivalence of diffusion models and EBMs allows us to consider sampling from the product of two different diffusion models $p_1(\tau)p_2(\tau)$, such that each diffusion model corresponds to an EBM, $e^{-E_1(\tau)}$ and $e^{-E_2(\tau)}$, and the product is given by $e^{-E'(\tau)} = e^{-(E_1(\tau) + E_2(\tau))}$. In particular, we can sample from this new distribution also by using Langevin sampling:

$$\tau_{t-1} = \alpha_t(\tau_t - \gamma \nabla_\tau E'_\theta(\tau_t) + \xi), \quad \xi \sim \mathcal{N}(\mathbf{0}, \sigma_t^2 \mathbf{I}), \tag{3}$$

which corresponds to the sampling procedure using denoising functions

$$\tau_{t-1} = \alpha_t(\tau_t - \gamma(\epsilon_{\theta_1}(\tau_t, t) + \epsilon_{\theta_2}(\tau_t, t)) + \xi), \quad \xi \sim \mathcal{N}(\mathbf{0}, \sigma_t^2 \mathbf{I}). \tag{4}$$

Below we will illustrate how this factored EBM parameterization of a diffusion model can allow a black-box pretrained model to be leveraged as a probabilistic prior.

## 3 PROBABILISTIC ADAPTATION OF BLACK-BOX TEXT-TO-VIDEO MODELS

To explain how a black-box text-conditioned video diffusion model can be effectively used as a probabilistic prior for video generation, we will first introduce the functional form of probabolistic adaptation in Section 3.1, and then discuss how the probabilistic composition can be implemented with diffusion models in Section 3.2. To generate high-quality videos, we also explain in Section 3.3 how the underlying probabilistic composition can be sharpened to generate low temperature samples.

### 3.1 BLACK-BOX TEXT-TO-VIDEO MODELS AS PROBABILISTIC PRIORS

Black-box text-to-video models were pretrained on massive datasets consisting of millions of videos, and are therefore able to capture a powerful prior $p_{\text{pretrained}}(\tau|\text{text})$ on the natural distribution of videos $\tau$. Given a smaller task-specific dataset of video-text pairs, $D_{\text{Adapt}} = \{(\tau_0, \text{text}_0), \ldots, (\tau_n, \text{text}_n)\}$,

Figure 2: **Video Adapter Framework.** Video Adapter only requires training a small domain-specific text-to-video model with orders of magnitude fewer parameters than a large video model pretrained from internet data. During sampling, Video Adapter composes the scores of the pretrained and the domain specific video models, achieving high-quality and flexible video synthesis.

how can one leverage the powerful prior captured by a pretrained video diffusion model to synthesize videos similar to those in $D_{\text{Adapt}}$? One approach is to directly finetune the weights of $p_{\text{pretrained}}(\tau|\text{text})$ using $D_{\text{Adapt}}$, but $p_{\text{pretrained}}(\tau|\text{text})$ has billions of parameters whose weights are often proprietary to private enterprises. Similar challenges with large language models have led to prompting and in-context learning. Analogously, we propose Video Adapter as a finetuning-free method to adapt pretrained video diffusion to a new dataset of videos $D_{\text{Adapt}}$ through probabilistic composition. Specifically, given $D_{\text{Adapt}}$, we learn a separate small video diffusion model $p_\theta(\tau|\text{text})$ to represent the distribution of videos in $D_{\text{Adapt}}$. We then adapt $p_{\text{pretrained}}(\tau|\text{text})$ to $D_{\text{Adapt}}$ by constructing a product distribution $p_{\text{produce}}(\tau|\text{text})$ in the form (see adaptation to multiple domains in Appendix D):

$$\underbrace{p_{\text{product}}(\tau|\text{text})}_{\text{Product Distribution}} \propto \underbrace{p_{\text{pretrained}}(\tau|\text{text})}_{\text{Pretrained Prior}} \underbrace{p_\theta(\tau|\text{text})}_{\text{Small Video Model}}, \tag{5}$$

By fixing the pretrained model $p_{\text{pretrained}}(\tau|\text{text})$, we train the small video model $p_\theta(\tau|\text{text})$ via maximum likelihood estimation on $D_{\text{Adapt}}$. This allows $p_\theta(\tau|\text{text})$ to exhibit high likelihood across videos in $D_{\text{Adapt}}$, but because $p_\theta(\tau|\text{text})$ is a small model trained on less diverse data, it can exhibit erroneously high likelihood across many unrealistic videos. The product distribution $p_{\text{product}}(\tau|\text{text})$ removes unrealistic videos by downweighting any videos $\tau$ that are not likely under the pretrained prior, enabling one to generate videos in the style of $D_{\text{Adapt}}$ that are realistic under $p_{\text{pretrained}}(\tau|\text{text})$.

## 3.2 IMPLEMENTING PROBABILISTIC ADAPTATION

To adapt the black-box model $p_{\text{product}}(\tau|\text{text})$ from Equation 5, as well as to sample from it, we exploit the EBM interpretation of diffusion models discussed in Section 2. Based on the EBM interpretation, the pretrained diffusion model $p_{\text{pretrained}}(\tau|\text{text})$ corresponds to an EBM $e^{-E_{\text{pretrained}}(\tau|\text{text})}$ while the smaller video model $p_\theta(\tau|\text{text})$ parameterizes an EBM $e^{-E_\theta(\tau|\text{text})}$. The product distribution then corresponds to:

$$p_{\text{product}}(\tau|\text{text}) \propto p_{\text{pretrained}}(\tau|\text{text})p_\theta(\tau|\text{text}) \propto e^{-(E_{\text{pretrained}}(\tau|\text{text})+E_\theta(\tau|\text{text}))} = e^{-E'(\tau|\text{text})},$$

which specifies a new EBM $E'(\tau)$ from the sum of energy functions of the component models. Substituting this EBM into Equation 3, we can sample from the product distribution $p_{\text{product}}(\tau|\text{text})$ through the diffusion sampling procedure:

$$\tau_{t-1} = \alpha_t(\tau_t - \gamma\nabla_\tau(E_{\text{pretrained}}(\tau_t|\text{text}) + E_\theta(\tau_t|\text{text})) + \xi), \quad \xi \sim \mathcal{N}(\mathbf{0}, \sigma_t^2\mathbf{I})$$

which corresponds to sampling from Equation 1 according to

$$\tau_{t-1} = \alpha_t(\tau_t - \gamma(\epsilon_{\text{pretrained}}(\tau_t, t|\text{text}) + \epsilon_\theta(\tau_t, t|\text{text})) + \xi), \quad \xi \sim \mathcal{N}(\mathbf{0}, \sigma_t^2\mathbf{I}).$$

Thus, to probabilistically adapt the pretrained black-box model to a new dataset $D_{\text{Adapt}}$, we can use the standard diffusion sampling procedure, but change the denoising prediction to the sum of predictions from both the black-box pretrained model and the task-specific small model. To control the strength of the pretrained prior in final video generation, we can introduce a weight term $\lambda$ to scale the pretrained distribution $p_{\text{pretrained}}^\lambda(\tau|\text{text})$, which corresponds to scaling the denoised prediction from $\epsilon_{\text{pretrained}}(\tau_t, t|\text{text})$ by a scalar $\lambda$

$$\epsilon(\tau_t, t|\text{text}) = \epsilon_\theta(\tau_t, t|\text{text}) + \lambda\epsilon_{\text{pretrained}}(\tau_t, t|\text{text}). \tag{6}$$

The combined model can be further refined by integrating multiple steps MCMC sampling between each diffusion noise distribution similar to (Du et al., 2023a; Sjöberg et al., 2023). Note that the prior strength $\lambda$ can be tunable or time-dependent, which we found to be useful in practice (Appendix B).

## 3.3 ADAPTING LOW TEMPERATURE SAMPLING

In practice, text conditioning in the denoising model $\epsilon(\tau_t, t|\text{text})$ from Equation 6 are often parametrized using classifier-free guidance (Ho & Salimans, 2022) to generate sharp images or videos conditioned on text while avoiding distractions from spurious likelihood modes of diffusion

---

**Algorithm 1** Sampling algorithm of Video Adapter

---
**Input:** Pretrained black-box model $\epsilon_{\text{pretrained}}(\tau, t|\text{text})$, inverse temperature $\omega$, prior strength $\lambda$.
Initialize sample $\tau_T \sim \mathcal{N}(\mathbf{0}, \boldsymbol{I})$
**for** $t = T, \ldots, 1$ **do**
    $\tilde{\epsilon}_{\text{text}} \leftarrow \epsilon_\theta(\tau_t, t|\text{text}) + \lambda\epsilon_{\text{pretrained}}(\tau_t, t|\text{text})$            // Compute score using text-conditioned prior.
    $\epsilon \leftarrow \epsilon_\theta(\tau_t, t)$                        // Compute unconditional score.
    $\tilde{\epsilon}_{\text{cfg}} \leftarrow \epsilon + \omega(\tilde{\epsilon}_{\text{text}} - \epsilon)$             // Compute weight for low temperature sampling.
    $\tau_{t-1} = \texttt{ddpm\_sample}(\tau_t, \tilde{\epsilon}_{\text{cfg}})$         // Run diffusion sampling (can use other samplers).
**end for**

---

models. This corresponds to sampling from the modified probability distribution:

$$p^{\text{cfg}}(\tau|\text{text}) \; \propto \; p(\tau)\left(\frac{p(\tau|\text{text})}{p(\tau)}\right)^\omega \; \propto \; p(\tau)p(\text{text}|\tau)^\omega,$$

where $\omega$ corresponds to the classifier free guidance strength, typically chosen to be significantly larger than 1. By upweighting the expression $p(\text{text}|\tau)$ via the inverse temperature $\omega$, the modified distribution $p^{\text{cfg}}(\tau|\text{text})$ above can generate lower temperature video samples conditioned on the text. It appears straightforward to similarly construct low temperature samples when adapting the black-box model by sampling from the distribution

$$p^{\text{cfg}}_{\text{product}}(\tau|\text{text}) \propto p^{\text{cfg}}_{\text{pretrained}}(\tau|\text{text})p^{\text{cfg}}_\theta(\tau|\text{text}), \tag{7}$$

but using the classifier-free distribution $p^{\text{cfg}}_{\text{pretrained}}(\tau|\text{text})$ as the probabilistic prior is now problematic, since classifier-free guidance has restricted $p^{\text{cfg}}_{\text{pretrained}}(\tau|\text{text})$ to very few high probability modes which might be incompatible with $D_{\text{Adapt}}$. To effectively leverage a broad probabilistic prior while simultaneously generating low temperature samples emulating $D_{\text{Adapt}}$, we propose to first construct a new text-conditioned video distribution following Section 3.1:

$$p_{\text{product}}(\tau|\text{text}) \propto p_{\text{pretrained}}(\tau|\text{text})p_\theta(\tau|\text{text}).$$

We can then use the density ratio of this composed text-conditioned distribution with the unconditional video density $p_\theta(\tau)$ learned on $D_{\text{Adapt}}$ to construct a new implicit classifier $p_{\text{product}}(\tau|\text{text})$. By increasing the inverse temperature $\omega$ on this implicit classifier, we can generate low temperature and high quality video samples conditioned on a given text by sampling from the modified distribution:

$$p^*_{\text{product}}(\tau|\text{text}) = p_\theta(\tau)\left(\frac{p_{\text{product}}(\tau|\text{text})}{p_\theta(\tau)}\right)^\omega,$$

which corresponds to sampling from a modified denoising function:

$$\tilde{\epsilon}_\theta(\tau, t|\text{text}) = \epsilon_\theta(\tau, t) + \omega(\epsilon_\theta(\tau, t|\text{text}) + \lambda\epsilon_{\text{pretrained}}(\tau, t|\text{text}) - \epsilon_\theta(\tau, t))$$

We quantitatively and qualitatively ablate the effect of this denoising function in Figure 6 and Table 2, showing that this variant leads to better blending of styles between models. The overall pseudocode for the proposed approach with classifier-free guidance is given in Algorithm 1.

## 4 EXPERIMENTS

In this section, we illustrate how a black-box pretrained text-to-video model can deliver a rich set of downstream capabilities when combined with a task-specific video model. In particular, leveraging a high quality and broad probabilistic prior enables (1) controllable video synthesis from edge-only inputs, (2) high-quality video modeling that outperforms both the pretrained model and the task-specific video model, and (3) domain randomization and data augmentation for robotics. See experiment details and additional experimental results in Appendix B and in supplementary material.[1]

### 4.1 ADAPTING TO SPECIFIC VIDEO DOMAINS

**Setup.** We first demonstrate that the probabilistic prior in Video Adapter can be used to adapt and modify the styles of videos. We curate two adaptation datasets $D_{\text{Adapt}}$, one with an "animation" style and the other with a "scifi" style, where videos containing relevant keywords in their descriptions are grouped together to form $D_{\text{Adapt}}$. A black-box large video diffusion model with 5.6B parameters was pretrained on mapping Sobel edges to all videos, and two task-specific small models with 330M parameters were trained to map Sobel edges to $D_{\text{Adapt}}$ videos.

**Stylizing Video Generation.** In Figure 4 and Figure 5, we demonstrate how the pretrained prior can adapt the animation and scifi models to alternative styles while maintaining the original animation and scifi contents. These results show that Video Adapter can effectively combine rich knowledge of

---

[1]See video visualizations in the supplementary zip file.

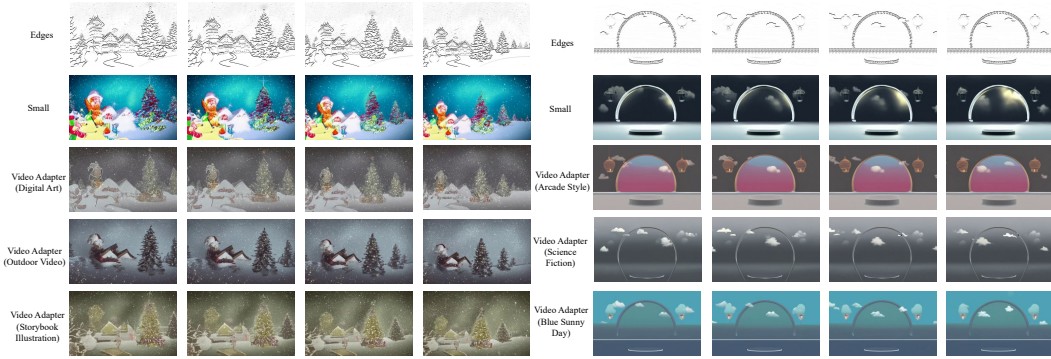

Figure 4: **Video Adapter enables stylization of a Animation Specific Model.** Video Adapter enables a large pretrained model to adapt and change the style a small animation style model.

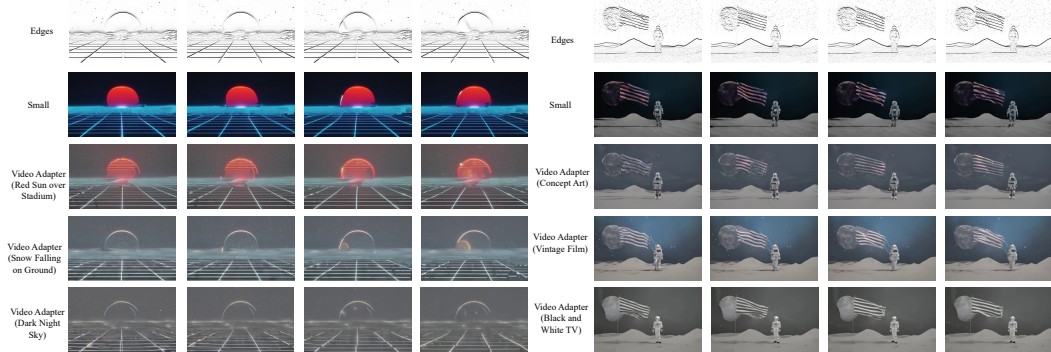

Figure 5: **Video Adapter enables stylization of a SciFi Specific Model.** Video Adapter enables a large pretrained model to adapt and change the style a small Scifi animation style model.

styles from the black-box model, such as "digital art", "outdoor video", "storybook illustration", with the animation content of the small model, thereby achieving flexible stylization.

**Specific Animation Style.** We further trained a small video model on an "animation" style of a particular artist. In Figure 3, we illustrate how the pretrained prior can maintain the anime content while changing the styles such as background color.

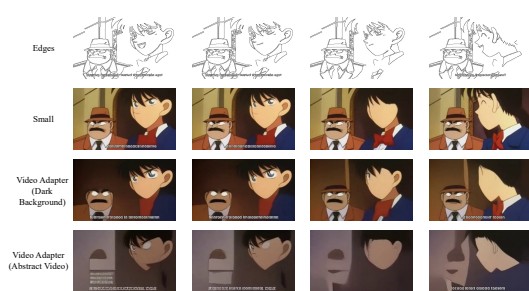

Figure 3: **Instance Specific Stylization.** Video Adapter enables the stylization of video model trained on a single animation style

**Analysis.** In Figure 6, we change the magnitude of the weight on the pretrained prior, and compare Video Adapter with directly interpolating the classifier-free scores between the pretrained and adapter models (as in Equation 7). We find that Video Adapter maintains the adapter style more accurately, whereas classifier-free score interpolation collapses to the teacher style with intermediate interpolation, leading to erratic artifacts.

## 4.2 HIGH-QUALITY EFFICIENT VIDEO MODELING

**Setup.** To demonstrate Video Adapter's ability in adapting the black-box pretrained model to domains that are not a part of pretraining, we consider adapting to Ego4D (Grauman et al., 2022) and Bridge Data (Ebert et al., 2021). These adaptations are nontrivial, as Ego4D consists of mostly egocentric videos that are not commonly found on the internet. Similarly, the Bridge Data consists of task-specific videos of a WidowX250 robot that is out of the distribution of the pretraining data. For Ego4D, we take a subset of the original dataset consisting of 97k text-video pairs and split them into train (90%) and test (10%) to form $D_{\text{Adapt}}$. For the Bridge Data, we take the entire dataset consisting of 7.2k text-video pairs and use the same train-test split to form $D_{\text{Adapt}}$.

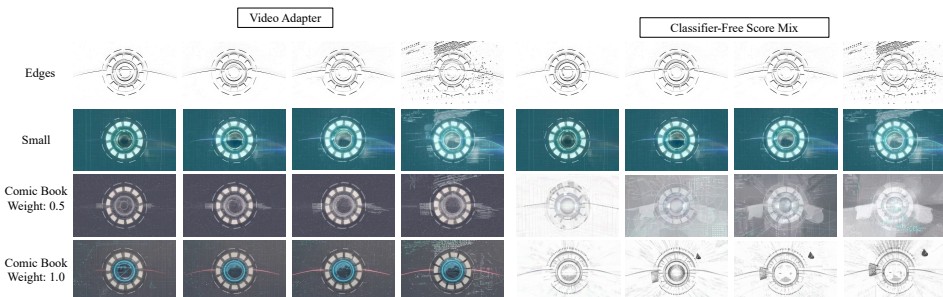

Figure 6: **Analysis of Video Adapter.** As adaptation weight increases, Video Adapter modifies the style as instructed (left), whereas directly mixing two classifier-free guidance scores fails to adapt the video (right).

| Model | Bridge | | | Ego4D | | |
|---|---|---|---|---|---|---|
| | FVD ↓ | FID ↓ | Param (B)↓ | FVD ↓ | IS ↑ | Param (B) ↓ |
| Small (S) | 186.8 | 38.8 | 0.07 | 228.3 | 2.28 | 0.07 |
| Small (S) + Pretrained | **177.4** | **37.6** | 0.07 | 156.3 | 2.82 | 0.07 |
| Small (L) | 152.5 | 30.1 | 0.14 | 65.1 | 3.31 | 2.8 |
| Small (L) + Pretrained | **148.1** | **29.5** | 0.14 | **52.5** | **3.53** | 2.8 |
| Pretrained | 350.1 | 42.6 | 5.6 | 91.7 | 3.12 | 5.6 |
| Pretrained Finetune | 321.0 | 39.4 | 5.6 | 75.5 | 3.33 | 5.6 |

Table 1: **Video Modeling Quantitative Performance** Video Adapter (Small + Pretrained) achieves better FVD, FID, and Inception Scores than both the pretrained model, pretrained model finetuned for equivalent number of TPU hours, and the task-specific small model with parameters as fewer as 1% of the pretrained model.

For the pretrained model, we use the 5.6B base model pretrained on generic internet videos from (Ho et al., 2022). For the task-specific small model, we downscale the video diffusion model from (Ho et al., 2022) by a factor of 80, 40, and 2 to create a diverse set of small models to be trained on task-specific $D_{\text{Adapt}}$. Table 1 shows the number of parameters of pretrained and small video models. Both the pretrained model and the small models are trained to generate subsequent frames conditioned on the first frame.

**Quantitative Results.** Table 1 shows the quantitative performance of Video Adapter under different video modeling metrics. On the Bridge Data, training a small model with parameters equivalent to 1.25% of the pretrained video model (first row) already achieves better metrics than the pretrained model. However, Video Adapter incorporating the pretrained model as a probablistic prior is able to further improve the metrics of the small model (second row). On Ego4D, due to the complexity of the egocentric videos, the smallest model with 1.25% of the pretrained video model can no longer achieve performance better than the pretrained model (first row), but incorporating the pretrained model during sampling still improves performance (second row). After increasing the size of the small model, Video Adapter is able to achive better metrics than both the pretrained and task-specific model. We further compare Video Adapter to finetuning the pretrained model for an equivalent number of TPU hours (see Appendix C), and show that Video Adapter achieves better performance than full tuning. Note that we only compare to full tuning out of curiosity as opposed to benchmarking, as the motivation of this work is the lack of weight access to the black-box pretrained models.

**Qualitative Results.** Figure 7 and Figure 8 show the generated videos on Bridge Data and Ego4D. On the Bridge Data in Figure 7, the pretrained model produces videos that do not correspond to the task described by the text (there is no robot arm movements in the generated video). The task-specific small model produces videos with unrealistic movements that teleport the robot arm. Video Adapter, on the other hand, produces videos with realistic movements that complete the task.

On Ego4D in Figure 8, the pretrained model produces high quality videos that contain little egocentric movement (first row), as the pretraining data mostly consists of generic videos from the internet that are not egocentric. The task-specific small model trained on Ego4D, on the other hand, produces videos with egocentric movement but of low quality (second row) due to limited model capacity. Video Adapter combines the best of both and generates high-quality egocentric videos (third row).

| Model | FVD ↓ | FID ↓ |
|---|---|---|
| CFG Mix | 167.4 | 33.1 |
| Small (L) | 152.5 | 30.1 |
| Video Adapter | **148.1** | **29.5** |

Table 2: **Ablations.** Video Adapter improves the underlying video modeling performance of models on while directly mixing classifier-free scores (CFG Mix) hurts performance.

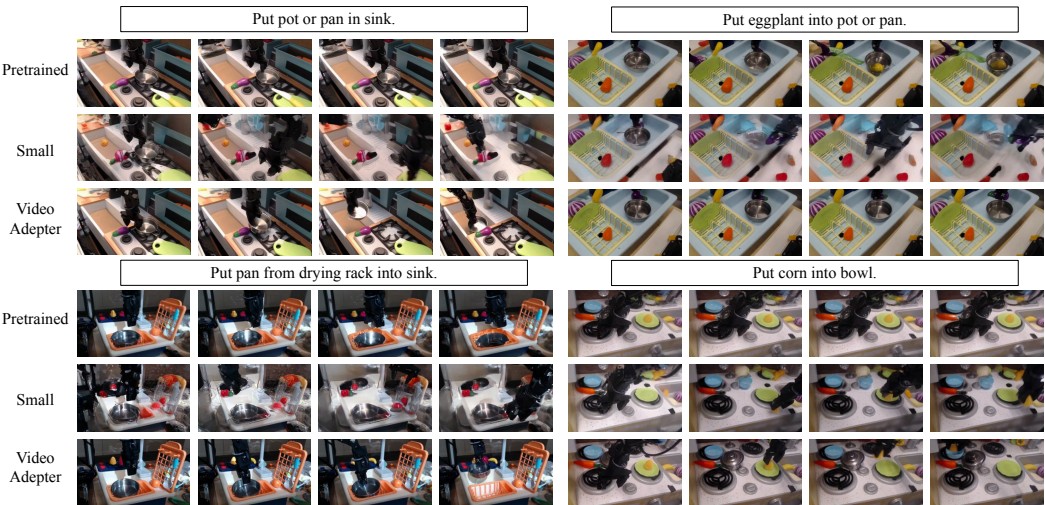

Figure 7: **Video Adapter on Bridge Data.** The pretrained model (first row) produces videos that are high-quality but are generally static and fail to complete the task. The small (L) model (second row) produces low-quality videos with unrealistic arm movements. Video Adapter (third row) produces high-quality videos and successfully completes the task.

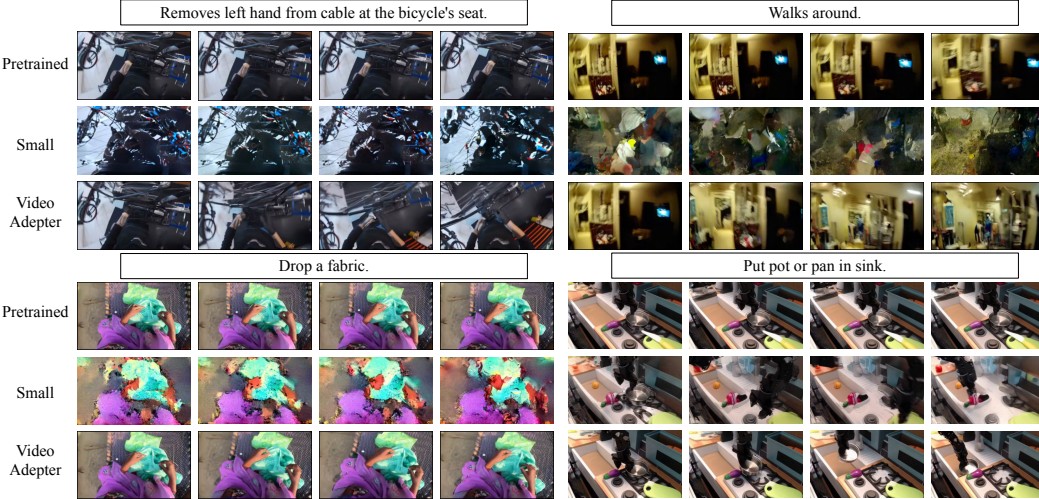

Figure 8: **Video Adapter on Ego4D.** The pretrained model (first row) produces high-quality but nearly static videos that do not reflect the egocentric nature.The small (L) model (second row) produces low-quality videos but with more egocentric movements. Video Adapter (third row) produces high-quality and egocentric videos.

**Ablations.** In Table 2, we report generative modeling performance of the small model on Bridge either using Video Adapter, or a interpolation between the classifier-free scores of pretrained and small models. We find that Video Adapter improves performance, while interpolation between classifier-free scores hurts performance.

### 4.3 SIM-TO-REAL VIDEO AUGMENTATION

**Setup.** One important application of controllable video synthesis is to render realistic robotic videos from simulation with a variety of data augmentations so that policies trained on the augmented observations are more likely to generalize well to real-world settings (Zhao et al., 2020). To demonstrate Video Adapter's capability in supporting sim-to-real transfer, we train a task-specific small edge-to-real model on 160k real robot trajectories of the LanguageTable dataset (Lynch et al., 2022), generating videos of execution conditioned on the Sobel edges of the real videos. Similarly, we train another small edge-to-sim model on 160k simulated robot videos. Note that the simulated and real robotics data are not paired (paired sim-to-real data are hard to find) but are connected through edge-conditioning. We again leverage the edge-conditioned large model pretrained on internet data for style specification.

Figure 9: **Video Adapter on sim-to-real transfer.** First row: simulated videos of execution plans generated by Video Adapter. Second row: real videos of execution plans generated by Video Adapter. Third row: real videos of execution plans generated by Video Adapter with data augmentation.

**Adapted Videos.** Figure 9 shows the generated robotic videos from Video Adapter. Video Adapter can effectively generate paired simulated and real robotic videos that complete a task described by a language prompt, and further generate videos with various data augmentation styles that can be utilized to train policies with better sim-to-real transfer abilities through techniques similar to domain randomization (Tobin et al., 2017).

## 5 RELATED WORK

**Text-to-Video Synthesis.** Following the recent success of text-to-image models (Nichol et al., 2021; Balaji et al., 2022; Ramesh et al., 2022; Rombach et al., 2022; Saharia et al., 2022b; Yu et al., 2022; Chang et al., 2023), large text-to-video models with autoregressive (Hong et al., 2022; Villegas et al., 2022; Wu et al., 2022a; 2021) and diffusion (Ho et al., 2022; Singer et al., 2022; Blattmann et al., 2023; Zhou et al., 2022; Esser et al., 2023) architectures have been developed, often by extending existing text-to-image models. Unfortunately, the model weights of large text-to-video models are generally not publically available, preventing downstream adaptations of these models.

**Adapting Pretrained Models** Adapting pretrained models for customized editing, inpainting, and stylization has been extensively studied in text-to-image and image-to-image translation models (Gal et al., 2022; Hertz et al., 2022; Kawar et al., 2022; Li et al., 2022; Lugmayr et al., 2022; Meng et al., 2021; Ruiz et al., 2022; Saharia et al., 2022a; Sasaki et al., 2021; Su et al., 2022). In text-to-video, most existing work either leverages text prompts (Molad et al., 2023; Esser et al., 2023), finetunes a pretrained model on stylized data (Wu et al., 2022b), or performing light training on a copy of the pretrained video model similar to ControlNet (Dhesikan & Rajmohan, 2023). Text-only adaption can be unreliable, whereas finetuning, prefix-tuning (Li & Liang, 2021), low-rank adaptation (Hu et al., 2021), and ControlNet all require access to the pretrained model weights, which are often not available for text-to-video models. Black-box adaptation has been applied extensively in language models (Brown et al., 2020; Wei et al., 2022; Liu et al., 2023), and large video models will soon face the same problem.

**Compositional Generative Models.** The techniques in this paper are further related to existing work on compositional generative modeling (Liu et al., 2022; Nie et al., 2021; Du et al., 2020; 2023a; 2021; Liu et al., 2021; Wang et al., 2023; Wu et al., 2022c; Deng et al., 2020; Urain et al., 2021; Gkanatsios et al., 2023; Gandikota et al., 2023; Po & Wetzstein, 2023), where different generative models are probabilistically combined to jointly generate outputs. In (Du et al., 2020), an approach to combine different probability distributions using EBMs is introduced. Most similar in spirit to this work, (Deng et al., 2020) composes a pretrained language model with a small EBM to improve language generation. However, different from this work, the small EBM is used to improve to global consistency of the language model, whereas we aim to use a small model to probabilistically adapt to a large pretrained video model to separate domains.

## 6 LIMITATION AND CONCLUSION

As video foundation models become more powerful but remain proprietary, black-box adaptation of these models is inevitible. We have proposed Video Adapter for leveraging black-box text-to-video models as probabilistic priors for guiding generation of specific videos. One limitation of Video Adapter is it still requires training a small domain-specific model, so adaptation is not completely training free. Another limitation is Video Adapter requires diffusion scores from pretrained black-box models. We advocate future video diffusion models to make scores as a part of the output to improve accessibility of these models.

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

# Appendix

In the Appendix we provided a detail derivation of connection between diffusion models and EBMs in Section A. We further provide additional experimental details in Section B. Finally, we provide a comparison with using the same computational budget to finetune the existing large pretrained model in Section C.

## A CONNECTION BETWEEN DIFFUSION AND EBM

The sampling procedure in a diffusion model corresponds to the Langevin sampling procedure on an EBM. To see this, we consider perturbing a sample $\tau^{t-1} \sim p\left(\tau^{t-1}\right)$ from target distribution $p(\tau^{t-1})$ with a Gaussian noise, *i.e.*,

$$\tau^t = \tau^{t-1} + \xi, \quad \xi \sim \mathcal{N}\left(\mathbf{0}, \sigma_t^2 \mathbf{I}\right)$$

which corresponds to the transition operator

$$\mathcal{T}(\tau^t | \tau^{t-1}) \propto \exp\left(\frac{-\left\|\tau^t - \tau^{t-1}\right\|^2}{2\sigma_t^2}\right)$$

where the joint distribution of $\tau^t$ and $\tau^{t-1}$ is

$$p(\tau^t, \tau^{t-1}) \propto \exp\left(\psi\left(\tau^{t-1}\right) - \frac{\left\|\tau^t - \tau^{t-1}\right\|^2}{2\sigma_t^2}\right).$$

We can express the Bayes estimator of $\tau^{t-1}$ given the perturbed observation $\tau^t$ as

$$m(\tau^t) = \int \tau^{t-1} p_\theta(\tau^{t-1} | \tau^t) d\tau^{t-1} = \tau^t + \sigma_t^2 \nabla \log p\left(\tau^t\right) \tag{8}$$

**Proof** By the property of Gaussian distribution, we have

$$\sigma^2 \nabla_{x'} p\left(x' | x\right) = p\left(x' | x\right)\left(x - x'\right). \tag{9}$$

Therefore, we have

$$\sigma \nabla_{x'} \int p\left(x' | x\right) p(x) dx = \int \left(x - x'\right) p\left(x', x\right) dx = \int x p\left(x', x\right) dx - x' p(x') \tag{10}$$

$$\Rightarrow \quad \sigma \nabla_{x'} \log p\left(x'\right) = \int x \frac{p\left(x', x\right)}{p\left(x'\right)} dx - x' = \mathbb{E}\left[X | x'\right] - x' \tag{11}$$

$$\blacksquare$$

Thus, we can represent the perturbed data with an EBM $p(\tau^t) \propto \exp\left(E_\theta\left(\tau^t, \sigma_t\right)\right)$, and learn the parameters through regression (Vincent, 2011; Saremi et al., 2018; Saremi & Hyvarinen, 2019; Song & Ermon, 2019), which leads to the optimal solution

$$\min_\theta \ \mathbb{E}_{\tau^{t-1} \sim \mathcal{D}, \xi \sim \mathcal{N}(0, \sigma_t^2 \mathbf{I})} \left[\left\|\tau^{t-1} - m(\tau^t)\right\|^2\right]$$

$$= \mathbb{E}_{\tau^{t-1} \sim \mathcal{D}, \xi \sim \mathcal{N}(0, \sigma^2 \mathbf{I})} \left[\left\|-\xi - \nabla E_\theta\left(\tau^{t-1} + \xi, \sigma_t\right)\right\|^2\right], \tag{12}$$

whch also corresponds to the denoising diffusion training objective.

Once we have the trained $E_\theta\left(\tau^t\right)$, we can then recover the sample $\tau^{t-1}$ according the denoising sampling procedure

$$\tau^{t-1} = \alpha^t m(\tau^t) + \alpha^t \xi = \alpha^t(\tau^t - \gamma \nabla_{\tau^t} E_\theta\left(\tau^t, \sigma_t\right)) + \alpha^t \xi, \quad \xi \sim \mathcal{N}\left(\mathbf{0}, \sigma_t^2 \mathbf{I}\right) \tag{13}$$

which corresponds to the sampling via stochastic localization (El Alaoui et al., 2022) and Equation 2 in the main paper.

## B EXPERIMENTAL DETAILS

### B.1 EXPERIMENT DETAILS

**Dataset** The large pretrained model is trained on 14 million video-text pairs plus 60 million image-text pairs, and with the LAION-400M image-text dataset. The images are spatially resized to 24x40 and videos using anti-aliased bi-linear resizing. We use different frame rate for different types of videos for best visualization results. For the Bridge (Ebert et al., 2021) we directly use the released opensource dataset. For Ego4D (Grauman et al., 2022) data, we take a small portion of the released dataset. For Anime and Sci-Fi style, we curate two separates datasets with their respective keywords. The keywords used for filtering data for Anime style are (in small letter) "disney", "cartoon", "anime",

"animation", "comic", "pixar", "animated", "fantasy". The keywords used for filtering data for Sci-Fi style are "science fiction", "sci-fi", "scifi", "astronaut", "alien", "NASA", "interstellar". For the animation with a particular artist style, we use the Case Closed animation (also named Detective Conan). For the Language Table dataset, we used the data from (Lynch et al., 2022).

| Dataset | Pretrain | Bridge | Ego4D | Anime | Sci-Fi | Case Closed | LangTable Sim | LangTable Real |
|---------|----------|--------|-------|-------|--------|-------------|---------------|----------------|
| # Train | 474M | 2.3k | 97k | 0.6M | 21k | 5k | 0.16M | 0.16M |

Table 3: **Training data size.** Number of text-video or text-image pairs used for training the pretrained large model and each of the small model. Training data for particular styles can be magnitude smaller than the pretraining dataset.

**Architecture.** To pretrain the large model, we use the same pretraining dataset, base architecture, and training setup as (Ho et al., 2022), with modifications of first-frame conditioning for Bridge and Ego4D, and edge conditioning for stylisation and sim-to-real. Specifically, the large model architecture consists of video U-Net with 3 residual blocks of 1024 base channels and channel multiplier [1, 2, 4], attention resolutions [6, 12, 24], attention head dimension 64, and conditioning embedding dimension 1024. To support first frame conditioning, we replicate the first frame across all future frame indices, and concatenate the replicated first frame channel-wise to the noisy data following (Du et al., 2023b). To support edge conditioning, we run a sobel edge detector and use gradient approximations in the x-direction as the conditional video, and concatenates these edge frames with noisy data similar to first-frame conditioning. The large model consists of 5.6 billion parameters in total. For the set of small models for adaptation, Ego4D Small (L) has 512 base channels in each of the residual blocks. Ego4D Small (S) and Bridge Small (S) have a single residual block with 32 base channels. Bridge Small (L) has a single residual block with 64 base channels. The set of stylisation models (animation, sci-fi, and particular anime style) have 3 residual blocks and 256 base channels. For illustrating the generated videos at a higher resolution, we train two additional spatial super resolution models 24x40 $\rightarrow$ 48x80 (1.4B) and 48x80 $\rightarrow$ 192x320 (1.2B). We additionally use T5-XXL (Raffel et al., 2020) to process input text prompts which consists of 4.6 billion parameters, which we omit from the parameter count as all large and small models require text embeddings.

**Training and Evaluation.** We train each of our video diffusion models for 2M steps using batch size 2048 with learning rate 1e-4 and 10k linear warmup steps. The large 5.6B pretrained model requires 512 TPU-v4 chips, whereas various small models require anywhere between 8 and 256 TPU-v4 chips depending on the size. We use noise schedule log SNR with range [-20, 20]. We use 128 samples and 1024 samples to compute the FVD, FID, and Inception Scores metric on Bridge and Ego4dD, respectively.

**Sampling.** All diffusion models are trained with 1000 timesteps of sampling. To generate videos, we combined scores from both pretrained models and adapter models for all timesteps except the last 100 timesteps. The last 100 timesteps capture high frequency information in an image, and we found better image quality if we did not combine scores in these timesteps. For simplicity, we use a pretrained neural strength of 0.2 for Ego4D and 0.1 for Bridge, and 0.4 for all animation datasets, but found additional gains when using large neural strength at earlier timesteps and smaller ones later.

## C    COMPARISON TO FINETUNING

To illustrate the computational efficiency of Video Adapter, we further compare video modeling metrics of Video Adapter to finetuning the pretrained model for an equivalent number of TPU time. Specifically, the pretrained model requires 512 TPU-v4 chips whereas the small model on Bridge data requires 8 TPU-v4 chips. The small Bridge model requires 100k steps to reach convergence, and hence we finetune the pretrained model for 100,000 / 64 = 1,560 steps. Video Adapter achieves better FVD and FID than finetuning the pretrained model for an equal number of TPU steps as shown in Table 1.

## D    COMPOSING MULTIPLE FACTORS

Given a set of $N$ separate datasets $\{D_i\}_{i=1:N}$ specifying a set of $N$ different styles of videos, we can also apply our probabilistic adaptation framework across these $N$ models by learning $N$ separate distributions $\{p_i(\tau|\text{text})\}_{i=1:N}$. We can directly sample from the product distribution

$$\underbrace{p_{\text{product}}(\tau|\text{text})}_{\text{Product Distribution}} \propto \underbrace{p_{\text{pretrained}}(\tau|\text{text})}_{\text{Pretrained Prior}} \underbrace{\prod_{i=1:N} p_i(\tau|\text{text})}_{\text{N Different Video Models}}, \tag{14}$$

which now assigns high likelihood to videos that exhibit each of the composed styles.

We can directly sample from this composed model using the modified composite denoising function

$$\tilde{\epsilon}_\theta(\tau, t|\text{text}) = \epsilon_\theta(\tau, t) + \omega \sum_{i=1}^{N} (\epsilon_i(\tau, t|\text{text}) + \lambda \epsilon_{\text{pretrained}}(\tau, t|\text{text}) - \epsilon_i(\tau, t)),$$

which simply corresponds to using a weighted average over the predictions of each of the $N$ models.

## E    COMPARISON TO PARAMETER EFFICIENT FINETUNING

The problem setting of Video Adapter is that pretrained model weights are *not* accessible. This scenario is common in large language models (e.g., GPT-4, Bard), and large text-to-video models are heading in the same direction. Parameter efficient finetuning (e.g., LoRA, null-text inversion, prefix-tuning) requires access to pretrained model weights, whereas Video Adapter does not. Therefore Video Adapter is not comparable to parameter efficient finetuning.

Nevertheless, we conducted comparisons to LoRA and null-text inversion out of curiosity (prefix-tuning is omitted since it has only been applied to language models). For LoRA, we use rank 1 and rank 64 to compare to the smaller and larger task-specific VideoAdapter model. For null-text inversion, we use an unconditional null embedding of size [64, 4096] (the same dimension as the original text embeddings). We report the video modeling metrics in Table 4.

| Model | Bridge | | Ego4D |
| | FVD ↓ | FID ↓ | FVD ↓ | IS ↑ |
|---|---|---|---|---|
| Small (S) | 186.8 | 38.8 | 228.3 | 2.28 |
| Small (S) + Pretrained | 177.4 | 37.6 | 156.3 | 2.82 |
| Small (L) | 152.5 | 30.1 | 65.1 | 3.31 |
| Small (L) + Pretrained | **148.1** | **29.5** | 52.5 | **3.53** |
| LoRA-Rank1 | 170.2 | 32.2 | 74.5 | 3.4 |
| LoRA-Rank64 | 165.5 | 31.6 | **50.3** | **3.5** |
| Null-text inversion | 288.8 | 40.2 | 90.2 | 3.1 |

Table 4: **Comparison of Video Adapter to parameter efficient finetuning** under fixed compute budget. Video Adapter performs close to LoRA-Rank64 on Ego4D and better than parameter efficient finetuning on Bridge.

We observe that LoRA-Rank1 performs slightly better than Video Adapter (small). However, In the LoRA-Rank1 case, LoRA still performs worse than training a small domain specific model. In this case, Video Adapter can simply use the small model without the pretrained prior. In comparison, we found LoRA-Rank64 leads to mixed results when compared to Video Adapter (large), i.e., LoRA outperforms Video Adapter on Ego4D but not on Bridge data. We found that null-text inversion performs the worst, potentially due to limited flexibility of null-embeddings during finetuning.

Our results illustrate that Video Adapter, despite requiring only black-bo adaptation without access to pretrained model weights performs better than Null-text inversion and very comparably to LoRA finetuning (with pretrained model weights).

