# OpenReview forum: "Probabilistic Adaptation of Black-Box Text-to-Video Models"
_ICLR.cc/2024/Conference — ICLR 2024 poster_

### Official Review · Reviewer_4TFV · 2023-10-30

**Soundness:** 3 good
**Presentation:** 3 good
**Contribution:** 3 good
**Rating:** 8
**Confidence:** 5

**Summary:**

This paper investigates how to adapt a pre-trained text-to-video model to a specific domain without modifying the pretrained model. The authors propose Video Adapter, which trained a task-specific small video model and leverage pre-trained models to guide generation during inference.

**Strengths:**

Since this framework only requires training a small domain-specific text-to-video model, the domain-specific model can be adapted to any pretrained models as long as their input and output definitions are same.

Experiments clearly demonstrate the effectiveness of the proposed method.

**Weaknesses:**

The authors assume pretrained model weights are not accessible. Table 1 and figure 9 verifies the authors’ claim. However, it is not clear how does the model perform compared to other methods, such as LoRA and prefix-tuning if pretrained model weights are accessible.

**Questions:**

Method in Sec 3 is not limit to text-to-video generation and should also be applicable to text-to-image generation. Have you considered other use cases?

Only two models (a pretrained model and a domain-specific model) are involved in derivation in Sec. 3. If we have more than 1 domain-specific model, can we easily extend this method to multiple domain-specific models?

I know the assumption here is that pretrained models are not accessible during training. But what if we can access the pre-trained models during training, will it outperform LoRA and prefix-finetuning?

---

> ### Author Response · Authors · 2023-11-20
> **Author Response**
>
> > Comparison to parameter efficient finetuning.
>
> The problem setting of VideoAdapter is that pretrained model weights are **NOT** accessible. This scenario is common in large language models (e.g., GPT-4, Bard), and large text-to-video models are heading in the same direction. Parameter efficient finetuning (e.g., LoRA, null-text inversion, prefix-tuning) requires access to pretrained model weights, whereas VideoAdaptor does not. Therefore VideoAdapter is not comparable to parameter efficient finetuning.
>
> Nevertheless, we conducted comparisons to LoRA and null-text inversion out of curiosity (prefix-tuning is omitted since it has only been applied to language models). For LoRA, we use rank 1 and rank 64 to compare to the smaller and larger task-specific VideoAdapter model. For null-text inversion, we use an unconditional null embedding of size [64, 4096] (the same dimension as the original text embeddings). We report the video modeling metrics in the following table:
>
>
> | Method | Bridge FVD &darr; | Bridge FID &darr; | Ego4D FVD &darr; | Ego4d IS &uarr; |
> | ----------- | ----------- | ----------- | ----------- | ----------- |
> | VideoAdapter Small | 177.4 | 37.6 | 156.3 | 2.8 |
> | LoRA-Rank1 | 170.2 | 32.2 | 74.5 | 3.4 |
> | Small (no adaption)  | 165.5 | 30.1 | 65.1 | 3.3 |
> | Video Adapter Large | 148.1 | 29.5 | 52.5 | 3.5 |
> | LoRA-Rank64 | 165.5 | 31.6 | 50.3 | 3.5 |
> | Null-text inversion | 288.8 | 40.2 | 90.2 | 3.1 |
>
> We observe that LoRA-Rank1 performs slightly better than VideoAapter (small). However, In the LoRA-Rank1 case, LoRA still performs worse than training a small domain specific model. In this case, VideoAdapter can simply use the small model without the pretrained prior. In comparison, we found LoRA-Rank64 leads to mixed results when compared to VideoAdaptor (large), i.e., LoRA outperforms VideoAdapter on Ego4D but not on Bridge data. We found that null-text inversion performs the worst, potentially due to limited flexibility of null-embeddings during finetuning.
>
> Our results illustrate that VideoAdapter, despite requiring only black-bo adaptation without access to pretrained model weights performs better than Null-text inversion and very comparably to LoRA finetuning (with pretrained model weights).
>
> > Use case in text-to-image adaptation.
>
> VideoAdapter is indeed applicable to the text-to-image setting, as text-to-image is a special case of text-to-video. We focused on the text-to-video setting because the pretrained text-to-video models are generally much bigger than pretrained text-to-image models, which requires even more efficient adaptation.
>
> > Composing multiple models.
>
> This is not only possible but in favor of VideoAdapter compared to other adaptation methods (e.g., efficient finetuning), since VideoAdapter can compose arbitrary black-box models at inference time. We provide an additional analysis on adaptation in Appendix D.

---

> > ### Comment · Reviewer_4TFV · 2023-11-22
> >
> > Authors solve all my concerns and I would increase my rating.

---

### Official Review · Reviewer_nNvZ · 2023-10-31

**Soundness:** 4 excellent
**Presentation:** 4 excellent
**Contribution:** 3 good
**Rating:** 6
**Confidence:** 5

**Summary:**

This paper proposed a new method called Video Adapter to adapt a large pretrained text-to-video diffusion model for new transfer and enviroment. Video Adapter trained a small domain-specific model to provide the score to guide the inversion trajectory of large-scale pretrained model. The proposed method was successful in many downstream tasks including animation, style transfer and egocentric modeling.

**Strengths:**

1. This paper is easy to follow, and the overall writing is good.
2. The proposed method, Video Adapter, is a simple yet effective way to guide the inversion trajectory to adapt for new scenarios.

**Weaknesses:**

1. My major concern is the lack of comparison with other baseline models like Lora, null-text inversion and prefix-tuning. Surely, these methods I mentioned above were designed for image adaptation but, correct me if I am wrong, it seems that they can be easily transplanted to the video diffusion model to serve as baselines models. Authors should explain more about why such comparison is not necessary.

2.In table 1, small+pretrained demonstrate the best quantitative performance on both Bridge and Ego4D. However, these numbers are very close to only small variant. Considering the poor generative quality of small model in Figure 7 and 8, it is questionable whether small+pretrained can generate high-quality videos consistently.

**Questions:**

1. The prior strength weight is constant during the sampling. It would be interesting to see if that the prior strength weight is changing with t could be helpful.

I would increase my rating if my concerns are properly addressed.

---

> ### Author Response · Authors · 2023-11-20
> **Author Response**
>
> > Comparison to parameter efficient finetuning.
>
> The problem setting of VideoAdapter is that pretrained model weights are **NOT** accessible. This scenario is common in large language models (e.g., GPT-4, Bard), and large text-to-video models are heading in the same direction. Parameter efficient finetuning (e.g., LoRA, null-text inversion, prefix-tuning) requires access to pretrained model weights, whereas VideoAdaptor does not. Therefore VideoAdapter is not comparable to parameter efficient finetuning.
>
> Nevertheless, we conducted comparisons to LoRA and null-text inversion out of curiosity (prefix-tuning is omitted since it has only been applied to language models). For LoRA, we use rank 1 and rank 64 to compare to the smaller and larger task-specific VideoAdapter model. For null-text inversion, we use an unconditional null embedding of size [64, 4096] (the same dimension as the original text embeddings). We report the video modeling metrics in the following table:
>
>
> | Method | Bridge FVD &darr; | Bridge FID &darr; | Ego4D FVD &darr; | Ego4d IS &uarr; |
> | ----------- | ----------- | ----------- | ----------- | ----------- |
> | VideoAdapter Small | 177.4 | 37.6 | 156.3 | 2.8 |
> | LoRA-Rank1 | 170.2 | 32.2 | 74.5 | 3.4 |
> | Small (no adaption)  | 165.5 | 30.1 | 65.1 | 3.3 |
> | Video Adapter Large | 148.1 | 29.5 | 52.5 | 3.5 |
> | LoRA-Rank64 | 165.5 | 31.6 | 50.3 | 3.5 |
> | Null-text inversion | 288.8 | 40.2 | 90.2 | 3.1 |
>
> We observe that LoRA-Rank1 performs slightly better than VideoAapter (small). However, In the LoRA-Rank1 case, LoRA still performs worse than training a small domain specific model. In this case, VideoAdapter can simply use the small model without the pretrained prior. In comparison, we found LoRA-Rank64 leads to mixed results when compared to VideoAdaptor (large), i.e., LoRA outperforms VideoAdapter on Ego4D but not on Bridge data. We found that null-text inversion performs the worst, potentially due to limited flexibility of null-embeddings during finetuning.
>
> Our results illustrate that VideoAdapter, despite requiring only black-bo adaptation without access to pretrained model weights performs better than Null-text inversion and very comparably to LoRA finetuning (with pretrained model weights).
>
> > Poor video quality of small model in Figure 7 and 8.
>
> We used the larger version of the small model, i.e., Small (L), in Figure 7 and Figure 8. We found that for harder dataset (e.g., Ego4D), too small of an adapter model fails to adapt to the target domain. The size of the adapter should in fact be determined by the complexity of the video domain one wishes to adapt to, and VideoAdapter enables such flexibility during adaptation. We also found that video modeling metrics sometimes do not faithfully reflect sample quality, especially in failing to reflect whether the generated videos corresponds to the given text instruction.
>
> > Tunable prior strength.
>
> We have indeed found tunable prior strength to be helpful. For instance, we set the prior strength to 0 in the last 100 timesteps of diffusion (last paragraph in Appendix B.1), which we found to result in better generation quality than a fixed prior strength. In general, we found that  higher strengths at higher noise levels were helpful – perhaps due to more accurate score functions at higher timesteps. We have made tunable prior strength more explicit in Section 3.2 (Page 4).

---

> > ### Author Response · Authors · 2023-11-22
> > **Author Follow-up**
> >
> > Dear Reviewer,
> >
> > We would like to ask if your questions and concerns have been addressed by our response, and if there is anything else preventing you from increasing your score. Please let us know, and thank you for your time.

---

> > ### Comment · Reviewer_nNvZ · 2023-11-22
> > **Reply to authors**
> >
> > The rebuttal addressed most of my concerns. Therefore I'd like increase my rating.
> > For the 2nd weakness I have asked in my review, recent results in stable video diffusion also demonstrate the ineffectiveness of FVD as it cannot properly evaluate whether a model generate authetic video frames. I think the numbers in Fig 7 and 8 are another example of this ineffectiveness

---

### Official Review · Reviewer_yyCL · 2023-11-01

**Soundness:** 3 good
**Presentation:** 3 good
**Contribution:** 3 good
**Rating:** 8
**Confidence:** 3

**Summary:**

The paper proposes an adapter-based video-generation model training methodology. They use a large pre-trained videogen model and use the output scores from the model (at each step of denoising in diffusion models) to help train a smaller model for a specific task.
The black box large pre-trained model has been used as a prior to learn a distribution of new stylistic videos in a finetuning-free, probabilistic composition framework. The small model is learned via MLE and the black box model EBM interpretation is used to sample from it. To probabilistically adopt the pre trained black-box model to new data, the denoising prediction is changed to the sum of predictions from both the black-box pre-trained model and the task-specific small model. The combined model is further refined by MCMC sampling.
 The authors show successful experiments on controlled video generation, egocentric videos, and realistic robotic videos from simulation.

**Strengths:**

1. The authors tackle a novel problem of building proprietary video models which has not been addressed before in the video generation domain. The videos provided as supplementary material look great.
2. The paper is well-written and self-explanatory with well-cited results and related work

**Weaknesses:**

The paper is based on the fact that there will be more closed source companies willing to open source their diffusion scores.

**Questions:**

The description of “Controllable video synthesis” is not very clear in terms of inputs. The text description as inputs is not provided very clearly hence someone without knowing the problem statement of this particular problem has difficulty in understanding what are the inputs.

---

> ### Author Response · Authors · 2023-11-20
> **Author Response**
>
> Thank you for your positive evaluation of the paper. We address each of the questions and concerns below.
>
> > Companies Willing to Opensource Diffusion Scores
>
> We agree that it is not entirely clear if companies would be willing to open-source predicted diffusion scores, but believe that there is no free lunch – only with more access to the generation process would we be able to more effectively adapt models. The aim of our paper is to demonstrate that with relatively black-box access (only access to the score predictions during generation) we can already achieve effective adaptation of diffusion models, motivating companies to enable access to score predictions. There is also some precedent to releasing some intermediate generation information from companies – LLM APIs for instance often release intermediate logit likelihoods.
>
> > Textual Clarification
>
> We have updated “Controllable video synthesis” to “Adapting to Specific Video Domains” for clarity.

---

### Official Review · Reviewer_Z2pV · 2023-11-01

**Soundness:** 2 fair
**Presentation:** 2 fair
**Contribution:** 1 poor
**Rating:** 3
**Confidence:** 4

**Summary:**

The Video Adapter framework addresses the challenge of adapting black-box, large text-to-video models to specific domains such as robotics and animation, without access to the models' weight parameters. This innovative approach, inspired by the prompting mechanism in large language models, uses the score function of a pretrained video diffusion model as a probabilistic prior, guiding the generation of a smaller, task-specific video model. Remarkably, this smaller model, with only 1.25% of the parameters of the original, can generate high-quality and domain-specific videos across diverse applications. The authors advocate for private enterprises to provide access to the scores of video diffusion models, enhancing the adaptability of large pretrained text-to-video models, with video demos available in the supplementary material.

**Strengths:**

1. The tackled problem is important and practical.
2. The proposed idea is intuitive.
3. Some of the results look interesting.

**Weaknesses:**

1. The current manuscript could really use a better position to avoid over-claiming.

a. Currently the paper highlights the ability of adapting black-box pretrained large text-video model. However, it relaxes the assumption of obtaining intermediate generation results of the black-box model, which is not realistic, and even contradicts with the term "black-box". It is really inappropriate to just use an existing term in a completely different setting.

b. The current manuscript also highlights parameter efficiency. However, the current evidence is not enough to really claim this. There has been some many parameter efficient fine-tuning techniques and the comparison with such existing method is really needed to claim superiority from this part. Specifically, comparisons with LoRA and other SOTA PEFT methods should be at least provided to help the audience really know the performance of existing methods.

c. The current setup is actually not that highlighting efficiency. The largest small model can even take half of the computation of the large model. It would be much more rigorous if the authors can set a computation budget and compare methods under a fixed budget.

2. Evaluation of the video generation. From qualitative inspect, I don't really think Video Adapter in Figure 8 is really better. Is there any case study further examining the human preference and the score obtained from automatic evaluation, especially on datasets like Ego4D?

**Questions:**

Please check weakness for details.

---

> ### Author Response · Authors · 2023-11-20
> **Author Response**
>
> Thank you for your constructive evaluation of the paper. We address each of the questions and concerns below.
>
> > Black Box Terminology.
>
> We believe that our approach does act in a “black box” manner since it does not require access or knowledge about any of the weights of a neural network (treating the neural network itself as a “black box”), but rather only access to predictions from the neural network (albeit at each point of sample generation). While this uses more information than having only the final generated sample, there is no free lunch, and the greater the information given, the more effectively models are able to be adapted. There is precedence for giving access to such intermediate predictions in black-box models, for example the chatGPT API gives intermediate prediction logits.
>
> However, if the reviewer feels strongly about this, we are happy to adopt another word to describe our approach.
>
> > Parameter Finetuning
>
> The problem setting of VideoAdapter is that pretrained model weights are NOT accessible. This scenario is common in large language models (e.g., GPT-4, Bard), and large text-to-video models are heading in the same direction. Parameter efficient finetuning (e.g., LoRA, null-text inversion, prefix-tuning) requires access to pretrained model weights, whereas VideoAdaptor does not. Therefore VideoAdapter is not comparable to parameter efficient finetuning.
>
> Nevertheless, we conducted comparisons to LoRA and null-text inversion out of curiosity (prefix-tuning is omitted since it has only been applied to language models). For LoRA, we use rank 1 and rank 64 to compare to the smaller and larger task-specific VideoAdapter model. For null-text inversion, we use an unconditional null embedding of size [64, 4096] (the same dimension as the original text embeddings). We report the video modeling metrics in the following table:
>
>
> | Method | Bridge FVD &darr; | Bridge FID &darr; | Ego4D FVD &darr; | Ego4d IS &uarr; |
> | ----------- | ----------- | ----------- | ----------- | ----------- |
> | VideoAdapter Small | 177.4 | 37.6 | 156.3 | 2.8 |
> | LoRA-Rank1 | 170.2 | 32.2 | 74.5 | 3.4 |
> | Small (no adaption)  | 165.5 | 30.1 | 65.1 | 3.3 |
> | Video Adapter Large | 148.1 | 29.5 | 52.5 | 3.5 |
> | LoRA-Rank64 | 165.5 | 31.6 | 50.3 | 3.5 |
> | Null-text inversion | 288.8 | 40.2 | 90.2 | 3.1 |
>
> We observe that LoRA-Rank1 performs slightly better than VideoAapter (small). However, In the LoRA-Rank1 case, LoRA still performs worse than training a small domain specific model. In this case, VideoAdapter can simply use the small model without the pretrained prior. In comparison, we found LoRA-Rank64 leads to mixed results when compared to VideoAdaptor (large), i.e., LoRA outperforms VideoAdapter on Ego4D but not on Bridge data. We found that null-text inversion performs the worst, potentially due to limited flexibility of null-embeddings during finetuning.
>
> Our results illustrate that VideoAdapter, despite requiring only black-bo adaptation without access to pretrained model weights performs better than Null-text inversion and very comparably to LoRA finetuning (with pretrained model weights).
>
> > Computational Efficiency.
>
> In the Bridge setting, our small model is substantially smaller in size than our original pretrained model. However, in more complex domains, it is necessary to use larger models to capture the characteristics of that domain, and thus our larger model is similar in size to our original pretrained model.  In the table above, however, we have equated the computational budget across both our approach and parameter-efficient adaptation methods and find that our approach performs comparably to such approaches.
>
> > Video Generation Qualitative Comparison
>
> We have added a video comparison in https://drive.google.com/file/d/16O_RsGhwDhUmUsHTf8lSW8LvvCbEQ01r/view?usp=sharing. From these video results, it’s clear that our approach substantially better than the pretrained model without adaptation.

---

> > ### Author Response · Authors · 2023-11-22
> > **Author Follow-up**
> >
> > Dear Reviewer,
> >
> > We would like to ask if your questions and concerns have been addressed by our response, and if there is anything else preventing you from increasing your score. Please let us know, and thank you for your time.

---

### Author Response · Authors · 2023-11-20
**Author Response Summary**

We thank all reviewers for their feedback. In response to Reviewer Z2pV, nNvZ, 4TFV, we added comparisons to parameter efficient finetuning (Appendix E). We observe that VideoAdapter performs comparable to LoRA under a fixed computation budget. However, we want to highlight that the major contribution of this work is to adapt pretrained models **WITHOUT** access to pretrained model parameters (unlike efficient finetuning). We view VideoAdapter as an important step towards bridging the gap between black-box large video foundation models in industry and academic research. We provide detailed response below and updated the manuscript according to reviewer feedback (updates highlighted in blue).

---

### Meta-Review · Area_Chair_q6ay · 2023-12-06

**Metareview:**

Based on the provided reviews and author responses, the decision to accept Submission4192 is justified. The paper proposes an innovative Video Adapter framework, a novel approach in the domain of text-to-video models. The methodology of adapting large, black-box pretrained text-to-video models to specific domains without needing access to the models' internal weight parameters is both unique and practical. Here is a summary of the ratings and feedback from the reviewers:

Reviewer Z2pV rated the paper poorly, raising concerns about the realism of the assumptions, the need for better comparisons with existing methods, and questioning the efficiency and effectiveness of the proposed approach. Despite the authors' attempt to address these concerns, the reviewer remained unconvinced. This reviewer gave a final rating of "reject."

Reviewer yyCL had a positive view of the paper, highlighting its novelty and the quality of supplementary materials. This reviewer expressed some reservations about the practicality of companies releasing diffusion scores but ultimately rated the paper as "good" for acceptance.

Reviewer nNvZ expressed concerns about the lack of comparison with baseline models and the quality of generated videos. However, the reviewer found the authors' rebuttal satisfactory and increased their rating to "marginally above the acceptance threshold."

Reviewer 4TFV rated the paper as good, appreciating the framework's adaptability and effectiveness demonstrated in experiments. The reviewer's concerns were addressed by the authors, maintaining a positive assessment.

**Justification For Why Not Higher Score:**

Reviewer Z2pV  remained unconvinced with the concerns below. However, on balance, the AC thinks that paper can still be accepted due to other strengths.

"The authors said LLM also provides "intermediate" logits. But this is fundamentally different from the intermediate generation results of the text-video. The "decoding" process from the logits of a LLM is non-parametric. However, this is not the case with text-video models. It is a strong requirement to have access to the intermediate results from text-video models. It is also unrealistic since it could significantly increase the communication overhead, which makes the proposed setting most "gray-box" or more accurately, without access to parameters.
There are also existing lines of work specifically tackling the black-box tuning problem of LLM that is completely missing from the current manuscript [a].
If the text-video models are latent diffusion models, this would even mean that the indeterminate latents are necessary. This is also another part that the current work overclaims: it only tackles and validates models operating in the pixel space but over-claims as adaptation for text-video models.
The computation budget issue is even larger. The authors mysteriously annotate that the budget of training the smallest variant of their model but their models have different sizes and it is unclear whether the computation budget is fair for all the models compared in Table 1 on two of the datasets.
The large computation required by their model on Ego4D also indicates that the current problem setting is not realistic. It basically means that the pretrained large model is not effective enough as LLMs do in NLP. Therefore, all the current conclusions may not hold when the large pretrained text-video model is strong enough.
The additional example is not convincing at all. I disagree that the results of Video-adapter is much better. They are just both bad. That's why I suggest studying automatic evaluation and human preference. I suspect the current evaluation does not faithfully reflect the actual human preference. However, the authors didn't provide such study.
[a] Black-Box Tuning for Language-Model-as-a-Service ICML 2022"

**Justification For Why Not Lower Score:**

The majority of reviewers recognized the paper's strengths in addressing a novel problem and its practical implications. Despite Reviewer Z2pV's rejection, they did not respond after the author's rebuttal. The positive ratings and constructive feedback from other reviewers, along with the authors' comprehensive responses to concerns, support the acceptance of this paper.

---

### Decision · Program_Chairs · 2024-01-16

Accept (poster)